# Awareness of anthrax disease and the knowledge of its transmission and symtoms identification: A cross sectional study among butchers in ile-ife

Sunday Charles Adeyemo[1]*, Eniola Dorcas Olabode[1], Folashade Yetunde Adeleke[2], Sunday Olakunle olarewaju[3], Calistus Adewale Akinleye[3], Blessing Ele Idris[1], Israel Abiodun Rabiu[1], Oluwafemi Obehi Are-Daniel[4], James Ebunoluwa Atolagbe[5], Raheem Omotayo Olaposi[1]

**1** Department of Public Health and Biomedical Sciences, Institut Superieur de Sante, Niamey, Niger Republic, **2** NHIS Unit, Osun State University Teaching Hospitals, Osogbo, Nigeria, **3** Community Medicine Department, Osun State University, Osogbo, Nigeria, **4** School of Nursing, Midwifery or Social work, Christ Church University, Canterbury, Canterbury, Kent, United Kingdom, **5** Public Health Department, Adeleke University, Ede, Nigeria

\* charlespatho@gmail.com

## Abstract

Anthrax is a zoonotic disease of public health significance as it has led to the morbidity and mortality of human and livestock. Anthrax is caused by *Bacillus anthracis* which causes contamination of soil and water. Due to the grazing nature of cattle, they are mainly affected by anthrax among other herbivores. Butchers, as frontline actors in the meat value chain, are at high occupational risk. However, there is a paucity of data on their awareness and knowledge of this disease in urban settings in Nigeria, a gap this study aims to address. The study was a cross sectional study among 380 respondents selected using multistage sampling technique. Data was collected using a pretested, semi-structured interviewer-administered questionnaire. Data collected were analyzed with the use of IBM Statistical Package for Service Solutions (SPSS) version 25 software. Descriptive analysis was done for all variables. Majority of the respondents (78.9%) were not aware of anthrax. More than half of the respondents (56.2%) agreed that anthrax can be transmitted through contaminated soil. Majority of the respondents (87.5%) who were aware of anthrax reported that there is relationship between anthrax and sudden death of animals. Majority (62.5%) said animals with anthrax die suddenly without illness and have dark un-clotted blood flow from their body orifices. Majority (75.0%) said that carcass of animals with anthrax do not get stiff. The study concluded that there is poor awareness about anthrax and its transmission among respondents. However, among respondents who were aware of anthrax, majority have good knowledge about identification of anthrax symptoms.

**Data availability statement:** All relevant data are available within the paper and its Supporting Information files.

**Funding:** The authors received no specific funding for this work.

**Competing interests:** The authors have declared that no competing interests exist.

## Background

Anthrax is a globally significant zoonotic disease caused by the spore-forming bacterium *Bacillus anthracis.* It poses a dual threat of human morbidity/mortality and substantial economic loss, primarily through reduced livestock trade and productivity [1]. The disease manifests in three clinical forms: cutaneous (95% of cases), gastrointestinal, and inhalation (the most severe). All forms can progress to fatal systemic infection [2].

Herbivores are the most susceptible hosts, acting as primary amplifiers of the bacterium. Their deaths and decomposition contaminate soil and water with highly resilient spores, creating long-term environmental reservoirs [3, 4]. Human transmission occurs mainly through contact with or consumption of infected animal products and carcasses. The incubation period varies: cutaneous (1–7 days), gastrointestinal (1–5 days), and inhalation (typically 1 week to 2 months, though can be shorter) [5]. While livestock often present with sudden death and bloody discharge from orifices, human symptoms are initially non-specific (fever, malaise) but progress according to the route of infection. Occupations with high exposure risk include butchery, herding, farming, and animal product processing [6].

Although controlled in developed nations, anthrax remains endemic in many developing regions. An estimated 20,000–100,000 human cases occur annually, with approximately 64 million livestock farmers at risk [7]. Recent African outbreaks highlight its persistent threat: Uganda (2018) reported 186 human cases and 721 livestock deaths [8], and Kenya has experienced recurrent outbreaks [1]. In Nigeria, the first confirmed animal case of the 2023 regional outbreak was reported in July on a farm in Niger State, with livestock exhibiting sudden death and bleeding from orifices without clotting [9,10]. This followed a June 2023 outbreak in Ghana. The Nigeria Centre for Disease Control (NCDC) subsequently issued alerts, noting that significant animal movement from northern to southern states during the Ileya festival could have spread infected livestock [11]. Osun State, a region of pastoral activity and inward migration of herders, received livestock during this festival. Its border with Kwara State and large Muslim population make it a potential zone for disease introduction. Ile-Ife, a major urban center in Osun State, is home to a large community of butchers at high occupational risk.

Early recognition of anthrax in animals is the cornerstone of preventing human outbreaks. Butchers, as critical gatekeepers in the meat supply chain, are a key frontline group. Their awareness of the disease as well as knowledge of disease symptoms and transmission routes directly impacts community-level prevention and reporting. This study therefore aims to assess the awareness of anthrax, knowledge of its transmission mechanisms, and clinical signs among butchers in Ile-Ife, Osun State. The findings will inform targeted public health interventions to mitigate this ongoing zoonotic threat.

## Methods

### Study design and setting

A descriptive cross-sectional study was conducted in February 2024 to assess the knowledge of anthrax among commercial butchers in Ile-Ife, Osun State,

**Global Public Health**
PLOS

south-western Nigeria. The study setting comprised the four Local Government Areas (LGAs) of Ife Central, Ife East, Ife North, and Ife South, and the Modakeke Area Office, which constitute the main metropolitan and peri-urban areas of Ile-Ife [12]. (Fig 1)

### Study population

The study population was all licensed commercial butchers with fixed stalls or stands in the designated area.

### Inclusion criteria

1) Being an active, licensed commercial butcher

2) Operating from a fixed stall/stand within one of the selected markets.

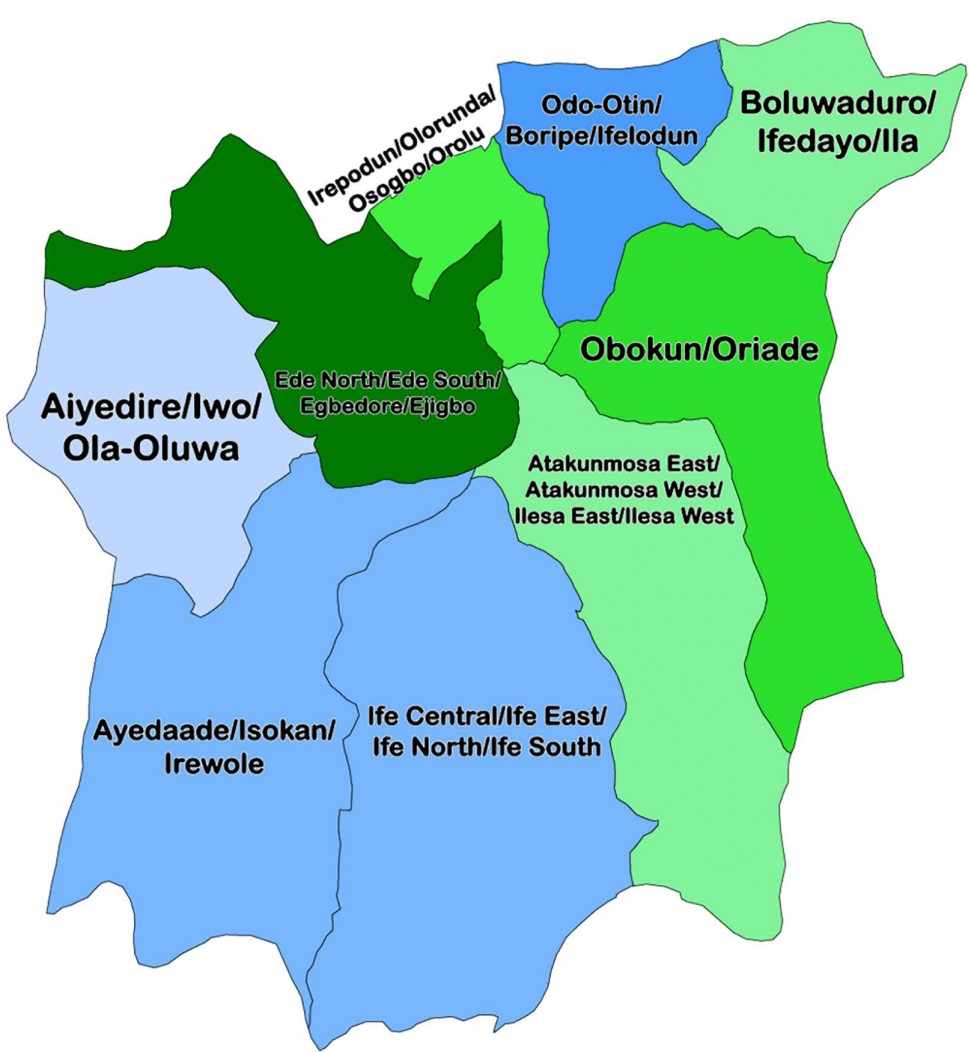

**Fig 1. Illustrates the geographical location of Ile-Ife, Osun State, Nigeria, where the study was conducted.** Available at https://commons.wikimedia.org/wiki/File:2022_Osun_State_gubernatorial_election_by_Federal_Constituency_margin.png.

## Exclusion criteria

1) Acute illness at the time of interview

2) Unwillingness to provide informed consent.

## Sample size determination

Sample size was calculated using Leslie Fischer's Formula $n = z^2pq/d^2$

z = Standard Normal Deviate which is taken as 1.96 @ 95% confidence interval

q = 1-p

p = Expected Outcome which was taken as 51.0% (0.51) which is the knowledge about human anthrax in Kisimu, Kenya [1].

d = Precision which was taken as 5% (0.05)

n= (1.96) *2 × 0.51 × 0.49/ (0.05) *2

= 0.960/ 0.0025 = 384

Adjusting for 10% non-response, an approximate of 450 respondents were recruited.

## Sampling techniques

A two-stage cluster sampling technique was employed.

Stage 1 – Selection of Markets (Primary Sampling Units): A comprehensive sampling frame of all major and minor markets in the five LGAs/Area Office was compiled from the Osun State Ministry of Trade and Investment, resulting in a list of 28 markets. Using a simple random sampling method (computer-generated random numbers), two markets were selected from each LGA/Area Office, culminating in a total of 10 markets.

Stage 2 – Selection of Butchers (Secondary Sampling Units): In each selected market, a site visit was conducted to obtain a verified list of all licensed commercial butchers (the secondary sampling frame) from the market leadership. From each list, butchers were selected using systematic random sampling. The sampling interval (k) was calculated for each market by dividing the total number of listed butchers by the required sample for that market (45 butchers per market, i.e., 450/10). Every k-th butcher was selected, starting from a number chosen by balloting between 1 and k.

## Data collection

A semi-structured, interviewer-administered questionnaire was developed in English and translated to Yoruba (the local language) by a certified linguist, followed by back-translation to ensure conceptual accuracy.

The questionnaire contained four sections:

1. Section A: Socio-demographic and occupational characteristics.

2. Section B: Awareness of Anthrax and source of information

3. Section C: Knowledge of anthrax transmission routes and clinical signs in animals and humans.

Validity and Reliability: Content validity was established through review by two veterinary public health experts and one epidemiologist. A pilot study was conducted with 45 butchers in a non-selected market in Ilesa, Osun State, to test for clarity, flow, and duration. The internal consistency of the knowledge section was assessed using Cronbach's alpha, yielding a coefficient of 0.78, indicating good reliability. Ambiguous items identified during the pilot were refined before the main survey. Data were collected from 12th to 17th February, 2024 after written informed consent was obtained from all participants.

### Data analysis

Data collected were sorted and analyzed with the use of IBM Statistical Package for Service Solutions (SPSS) version 25 software. Descriptive statistics was used to summarize the demographic characteristics and knowledge levels of respondents.

### Scoring and categorization of knowledge

A knowledge scoring system was applied to Section C. Each correct answer was awarded 1 point, and incorrect or "I don't know" responses received 0 points. The total possible score was 10. The mean score of the study population was calculated. Respondents scoring above the mean were categorized as having "Good knowledge," while those scoring at or below the mean were categorized as having "Poor knowledge."

### Ethical approval and consent to participate

This study was approved by the Research Ethics Committee of the Osun State University, Osogbo (UNIOSUNHREC 2024/002).This study was conducted according to the Declaration of Helsinki for Medical Research involving Human Subjects.

Right of decline/withdraw from study: Respondents were told that participation is voluntary and they will not suffer any consequences if they chose not to participate.

Confidentiality of data: All information gathered were kept confidential and participants were identified using serial numbers.

Consent form: A written informed consent was obtained from all participants.

Non-maleficence: No harm is intended nor befell any respondent in the course of the research study. Respondents were reassured of this.

## Results

Questionnaire was administered to 450 respondents, however, only 380 questionnaire were correctly filled. Giving a response rate of 84.4%.

Table 1 shows the demographic characteristics of respondents. More than half (53.9%) of the respondents were aged between 20–35 years. Majority of the respondents (71.1%) had secondary education. Two hundred and thirty (60.5%) of the respondents were Muslims while almost all respondents (92.1%) were from Yoruba tribe. About half (47.4%) of the respondents earn between 41000 – 100000 naira monthly. Majority of the respondents (88.2%) were male. Majority of the respondents (96.1%) handle (either rear or slaughter) cow and more than half of the respondents (69.7%) buy animal from the market and slaughter the animals themselves.

### Awareness of anthrax

Result revealed that only 80 (21.1%) of the respondents have heard about Anthrax. The source of information for those who have heard about anthrax include friends/neighbours, mass/social media and local market in which half of the respondents got the information from local market. (Fig 2)

### Knowledge of anthrax transmission and symptoms

Out of the respondents who have heard about anthrax, only 18.8% agreed that anthrax can be transmitted from animal to man. Less than half (37.5%) agreed that anthrax can be transmitted by handling infected animal. However, more than half of the respondents (56.2%) agreed that anthrax can be transmitted through contaminated soil. (Table 2) Majority of the respondents (87.5%) who are aware of anthrax reported that there is relationship between anthrax and sudden death of

**Table 1. Sociodemographic characteristics of butchers interviewed in Ile-Ife, Osun State, Nigeria (N = 380).**

| Demographic characteristics | Frequency | Percentage (%) |
|---|---|---|
| **Age** | | |
| 20-35 | 205 | 53.9 |
| 36-45 | 125 | 32.9 |
| >45 | 50 | 13.2 |
| **Level of education** | | |
| Primary | 90 | 23.7 |
| Secondary | 270 | 71.1 |
| Tertiary | 20 | 5.3 |
| **Religion** | | |
| Christian | 150 | 39.5 |
| Muslim | 230 | 60.5 |
| **Tribe** | | |
| Yoruba | 350 | 92.1 |
| Hausa | 30 | 7.9 |
| **Income (in naira)** | | |
| 10000-20000 | 40 | 10.5 |
| 21000-40000 | 116 | 30.5 |
| 41000-100000 | 180 | 47.4 |
| >100000 | 44 | 11.6 |
| **Gender** | | |
| Male | 335 | 88.2 |
| Female | 45 | 11.8 |
| **Specie of livestock handled** | | |
| Cow | 365 | 96.1 |
| Goat | 10 | 2.6 |
| Pig | 5 | 1.3 |
| **Animal source** | | |
| Personal farm | 35 | 9.2 |
| Bought animal from market | 265 | 69.7 |
| Bought slaughtered animal from slaughter house | 80 | 21.1 |

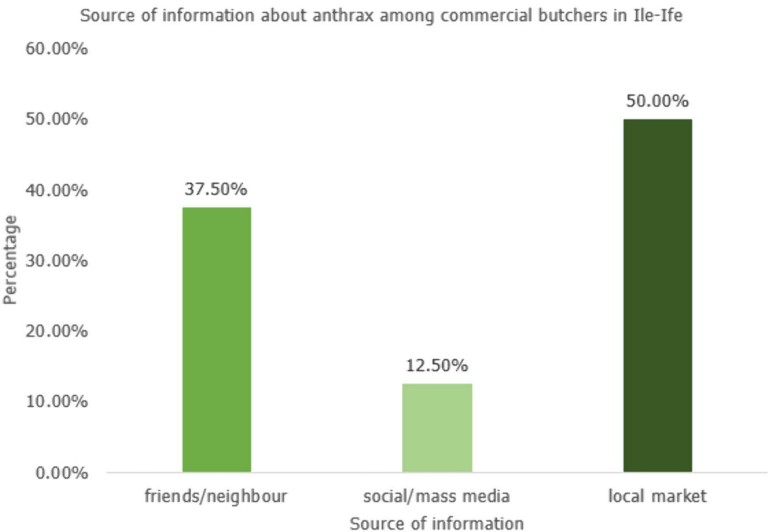

**Fig 2. Shows the distribution of respondents according to their reported sources of information about anthrax.**

PLOS Global Public Health

**Table 2. Knowledge about transmission of Anthrax among butchers interviewed in Ile-Ife, Osun State, Nigeria'.**

| Knowledge about transmission of Anthrax | Frequency | Percentage (%) |
|---|---|---|
| **Can anthrax be transmitted from animal to man (n=80)**<br>Yes<br>No | <br>15<br>65 | <br>18.8<br>81.2 |
| **Anthrax can be transmitted by handling infected animal (n=80)**<br>Yes<br>No | <br>30<br>50 | <br>37.5<br>62.5 |
| **Anthrax can be transmitted through contaminated soil (n=80)**<br>Yes<br>No | <br>45<br>35 | <br>56.2<br>43.8 |

animal. Majority (62.5%) said animals with anthrax die suddenly without illness and have dark un-clotted blood flow from their body orifices. Majority (75.0%) said that carcass of animals with anthrax do not get stiff. (Table 3).

Overall, based on the composite scores, 38 **(47.5%)** of the respondents demonstrated good overall knowledge of anthrax transmission and symptoms, while 42 **(52.5%)** demonstrated poor knowledge.

## Association between sociodemographic characteristics and knowledge of anthrax

The cross-tabulation analysis revealed significant associations between key sociodemographic factors and overall anthrax knowledge level among aware respondents. A respondent's level of education was significantly associated with knowledge ($\chi^2=8.92$, p=0.012), with those possessing tertiary education disproportionately represented in the "good knowledge" category. Similarly, monthly income showed a significant relationship ($\chi^2=6.15$, p=0.046), as individuals in the lowest income bracket (≤₦40,000) were markedly overrepresented in the "poor knowledge" group. The source of animals was also a significant factor ($\chi^2=7.84$, p=0.020); butchers who sourced livestock from their personal farms were far more likely to have good knowledge compared to those who purchased animals from the market. In contrast, no statistically significant associations were found between knowledge level and age group ($\chi^2=0.87$, p=0.647) or religion ($\chi^2=0.06$, p=0.806).

To control for potential confounding, a binary logistic regression was performed with overall knowledge (good/poor) as the dependent variable. The independent variables entered into the model were education, income, and source of animals. After adjustment, only educational attainment remained a statistically significant predictor of good knowledge. Butchers with tertiary education were over nine times more likely to have good knowledge compared to those with only primary education (AOR=9.25, 95% CI: 1.80-47.52, p=0.008). Income and source of animals were no longer significant in the adjusted model, suggesting that their effect in the bivariate analysis was confounded by other variables, particularly education.

## Discussion

This study assessed the awareness and knowledge of anthrax among butchers in Ile-Ife, a region at potential risk due to livestock trade patterns. The findings reveal a critical public health vulnerability, characterized by extremely low general awareness, significant gaps in essential knowledge among those aware, and dangerous misconceptions regarding transmission, a pattern that aligns with and underscores persistent challenges documented across West Africa.

The most striking finding is the profoundly low level of general awareness, with only 21.1% of all surveyed butchers having heard of anthrax. This prevalence is markedly lower than the 73% awareness reported among livestock farmers in northern Ghana but is consistent with the 26.3% found among pastoralists in the same region, indicating that knowledge is not only geographically heterogeneous but also critically low among key value-chain actors like butchers [4]. This deficiency is alarming, as basic awareness is the foundational step for any preventive behavior. The primary source of information for

**Table 3. Knowledge about identification of anthrax symptoms among butchers interviewed in Ile-Ife, Osun State, Nigeria.**

| Symptoms of Anthrax (n = 80) | Frequency | Percentage (%) |
|---|---|---|
| **Relationship between anthrax and sudden death of animal** | | |
| Yes | 70 | 87.5 |
| No | 10 | 12.5 |
| **Animal dies suddenly without illness sign** | | |
| Yes | 30 | 37.5 |
| No | 50 | 62.5 |
| **Dark un-clotted blood flow** | | |
| Yes | 30 | 37.5 |
| No | 50 | 62.5 |
| **Carcass does not become stiff** | | |
| Yes | 60 | 75.0 |
| No | 20 | 25.0 |

most aware respondents was the local market (50%), followed by friends and neighbours. This highlights the critical role of informal, community-based channels—and the concurrent risk of misinformation—over structured public health campaigns, a trend also observed in Cameroon where community networks were the primary information source for anthrax [13].

Among the minority who were aware of anthrax, knowledge of its zoonotic nature was critically poor. Only 18.8% correctly identified that anthrax can be transmitted from animals to humans. This represents a severe occupational blind spot and is considerably lower than the 46.1% of healthcare workers in Tanzania who recognized its zoonotic potential [14]. Furthermore, only 37.5% recognized transmission through handling infected animals, indicating that even those who know of the disease may not understand their specific risk pathway. This gap mirrors findings from Uganda, where communities with frequent animal contact had poor knowledge of specific transmission routes [15].

Knowledge of clinical signs in animals was inconsistent and incomplete. While a high proportion (87.5%) correctly associated anthrax with sudden death, a sign also widely recognized in studies in Ghana and Zambia, far fewer identified the specific signs such as sudden death without prior illness (37.5%) and dark, unclotted blood from orifices (37.5%) [16, 17]. This pattern suggests that butchers may recognize "sudden death" as a general problem but lack the precise diagnostic criteria to distinguish anthrax from other causes of acute mortality. The best-known sign was the lack of carcass stiffness (75.0%), possibly because it is a more tangible and observable post-mortem feature.

The overall knowledge classification found that less than half (47.5%) of the aware butchers had "good" knowledge. When viewed against the backdrop that only 21.1% of the total sample was aware, this translates to an approximate of 10% of all surveyed butchers possessing adequate knowledge. This profound deficit is a significant barrier to Nigeria's "One Health" outbreak response strategy. The recent 2023 outbreaks in Ghana and Nigeria underscore the urgency of this issue, as early detection by frontline actors is crucial for containment [9, 10].

The inferential analyses provide crucial nuance, identifying specific high-risk subgroups for targeted intervention. The bivariate analysis identified significant associations between good knowledge and higher education, higher income, and sourcing animals from a personal farm. However, the logistic regression analysis, which controlled for confounding, revealed that only higher educational attainment remained a significant independent predictor. Butchers with tertiary education were over nine times more likely to have good knowledge than those with only primary education. This underscores the fundamental role of formal education in enhancing the capacity to access, process, and utilize complex health information, a finding consistent with global health literacy models and studies from Ethiopia [6]. The fact that income and animal source were no longer significant after adjustment suggests that their apparent effect in the bivariate analysis was largely explained by their correlation with education. Individuals with higher incomes and those who own farms may also be more likely to be educated. This highlights education as the primary and most robust driver of knowledge, and the most

critical target for intervention. The lack of association between knowledge and age or religion suggests that these factors are not primary drivers of knowledge disparities within this occupational group.

Limitations of this study include its cross-sectional design, which captures knowledge at a single point in time, and its restriction to Ile-Ife, which may limit generalizability. Furthermore, knowledge was self-reported, which may be subject to social desirability bias.

## Conclusion

In conclusion, the severe deficit in anthrax awareness and knowledge among butchers in Ile-Ife mirrors critical gaps identified in recent studies across the continent, representing a tangible risk for undetected outbreaks. The findings reinforce the need for occupation-specific, context-adapted interventions.

We therefore recommend:

1. Targeted, participatory health education campaigns developed in collaboration with butchers' associations, leveraging the trusted local market channels identified in this and other studies.

2. Integration of zoonotic disease modules into informal apprenticeship programs and formal veterinary public health extension services.

3. Strengthening of surveillance systems by establishing clear, low-barrier reporting protocols for butchers, as successful models in other regions demonstrate.

Addressing these knowledge gaps is a critical component of regional health security, essential for preventing the morbidity, mortality, and economic loss associated with anthrax outbreaks.

## Supporting information

**S1 Table. Sociodemographic Variables.**
(DOCX)

**S2 Table. Animal & Exposure Variables.**
(DOCX)

**S3 Table. Behavioral & Knowledge Variables.**
(DOCX)

**S1 Data. Anthrax data.**
(SAV)

## Acknowledgments

The authors wish to acknowledge the respondents and the spouses of the authors for their understanding during the course of this study.

## Author contributions

**Conceptualization:** Sunday Charles Adeyemo, Eniola Dorcas Olabode.

**Data curation:** Folashade Yetunde Adeleke, Blessing Ele Idris, Israel Abiodun Rabiu, Raheem Omotayo Olaposi.

**Formal analysis:** Eniola Dorcas Olabode, Oluwafemi Obehi Are-Daniel.

**Investigation:** Folashade Yetunde Adeleke, Blessing Ele Idris, Israel Abiodun Rabiu, Raheem Omotayo Olaposi.

**Methodology:** Sunday Charles Adeyemo, Calistus Adewale Akinleye, Oluwafemi Obehi Are-Daniel.

**Project administration:** Sunday Charles Adeyemo, Eniola Dorcas Olabode, Folashade Yetunde Adeleke, Sunday Olakunle olarewaju, Calistus Adewale Akinleye, James Ebunoluwa Atolagbe.

**Supervision:** Sunday Charles Adeyemo, Sunday Olakunle olarewaju, Calistus Adewale Akinleye, James Ebunoluwa Atolagbe.

**Visualization:** Oluwafemi Obehi Are-Daniel.

**Writing – original draft:** Eniola Dorcas Olabode.

**Writing – review & editing:** Sunday Charles Adeyemo, Sunday Olakunle olarewaju.

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
