## [Decision Letter · Decision Letter 0]

30 Dec 2025

PGPH-D-25-03087

AWARENESS OF ANTHRAX DISEASE AND THE KNOWLEDGE OF ITS TRANSMISSION AND SYMPTOMS IDENTIFICATION: A CROSS SECTIONAL STUDY AMONG BUTCHERS IN ILE-IFE

Dear Dr. Adeyemo,

Thank you for submitting your manuscript to PLOS Global Public Health. After careful consideration, we feel that it has merit but does not fully meet PLOS Global Public Health’s publication criteria as it currently stands. Therefore, we invite you to submit a revised version of the manuscript that addresses the points raised during the review process.

The manuscript has been evaluated by two reviewers, and their comments are available below.

The reviewers have raised a number of concerns that need attention. In particular, they request additional information on methodological aspects of the study, additional analysis, a clearly defined research question, and improvements to the quality of the language used.

Could you please revise the manuscript to carefully address the concerns raised?

We look forward to receiving your revised manuscript.

Kind regards,

Helen Howard

Staff Editor

Journal Requirements:

1. Please amend your online Financial Disclosure statement. If you did not receive any funding for this study, please simply state: “The authors received no specific funding for this work.”

2. Please update your online Competing Interests statement. If you have no competing interests to declare, please state: “The authors have declared that no competing interests exist.”

3. We note that your Data Availability Statement is currently as follows: “Data for the study has been presented in the nanuscript”

Please confirm at this time whether or not your submission contains all raw data required to replicate the results of your study. Authors must share the “minimal data set” for their submission. PLOS defines the minimal data set to consist of the data required to replicate all study findings reported in the article, as well as related metadata and methods (https://journals.plos.org/globalpublichealth/s/data-availability#loc-minimal-data-set-definition).

If your submission does not contain these data, please either upload them as Supporting Information files or deposit them to a stable, public repository and provide us with the relevant URLs, DOIs, or accession numbers. For a list of recommended repositories, please see https://journals.plos.org/globalpublichealth/s/recommended-repositories.

4. Please provide separate main figure files in .tif or .eps format only and remove any figures embedded in your manuscript file. Please also ensure that all files are under our size limit of 10MB. Please leave the figure captions in the manuscript.

Additional Editor Comments (if provided):

Reviewers' comments:

Reviewer's Responses to Questions

**Comments to the Author**

1. Does this manuscript meet PLOS Global Public Health’s publication criteria?

Reviewer #1: No

Reviewer #2: Yes

2. Has the statistical analysis been performed appropriately and rigorously?

Reviewer #1: No

Reviewer #2: No

3. Have the authors made all data underlying the findings in their manuscript fully available (please refer to the Data Availability Statement at the start of the manuscript PDF file)?

Reviewer #1: No

Reviewer #2: No

4. Is the manuscript presented in an intelligible fashion and written in standard English?

Reviewer #1: No

Reviewer #2: No

Reviewer #1: I reviewed the manuscript titled ‘Awareness of anthrax disease and the knowledge of its transmission and symptoms identification: a cross sectional study among butchers in Ile-Ife’ and below are my observations:

Abstract

The authors were not consistent in the use of tenses. They should stick to the use of past tenses or continuous tenses. In line 43 they mentioned ‘were not aware’ but in 45 used ‘are aware’. Aware ‘of’ is more preferable than aware ‘about’. They were also not consistent in the use of singular or plural words. For instance, in line 46, they referred to ‘animal’ and then ‘animals’. They can stick to the former as a collective noun or the latter as plural.

The authors’ conclusion that there was poor awareness is not consistent with the fact that there was no mention of scoring the questionnaire in the methods. In addition, the lack of mention of knowledge which is a part of the title in the abstract leaves a major void. Knowledge should be captured both in the methods and results. Otherwise, it should be deleted from the title.

Introduction

Remove the space between the punctuation and the citation in square bracket see lines 63, 79 and 112, for instance. Anthrax is a disease (not an agent) caused by Bacillus anthracis not ‘Bacillus Anthrax’.

Line 75: Change ‘buy’ with ‘but’.

Authors should stick to ‘human or ‘humans’ (Lines 74 and 76).

The transition from one paragraph to the other is not smooth. For instance, lines 79 and 80 do not have any connection. In fact paragraph 4 has no place in its current position in the background of the study. The first sentence of paragraph 5 makes no sense and should be recast.

The statement that the first case of anthrax in Nigeria was in 2023 is a lie!

The authors should make use of the correct words: For instance, ‘body openings’ should be referred to as ‘body orifices’.

Line 104: The authors seem to have changed from reporting a study to writing a proposal.

Lines 105 – 108 require referencing.

Lines 111 and 112: Change ‘has’ to ‘have’.

Line 118: ‘Osun - State, South-Western’, should be written ‘Osun State, southwestern’.

Line 121: Change ‘it's’ to ‘its’.

Line 125: Change ‘accommodates’ to ‘accommodate’.

The authors should draw a map of the area being described.

Line 134: Change ‘accommodates’ to ‘accommodate’.

The inclusion and exclusion criteria are confusing: ‘Butchers who were standing were included’. Does this mean that those butchers who were sitting were excluded? ‘Butchers who were sick were excluded’, therefore only the healthy were included. How did the authors identify healthy and sick butchers? What was the reason for the chosen criteria?

Lines 140 - 149: The fact that the authors are contemplating calculating sample size and recruiting respondents for a study that is concluded is absurd.

Lines 158 and 159: Change ‘was’ to ‘were’.

Line 161: Which random sampling did the authors use to select the butchers? They should be specific in explaining how the sampling method used was applied in the work.

The sampling was not well explained. The authors claimed to have used multi-stage sampling technique. What was the sampling frame in each stage and how were they prepared or accessed?

The authors did not explain whether and how the questionnaire was validity and reliability tested. Was there a pilot study and how was it conducted? Without these, the instrument is not valid for a meaningful research. How many sections did the instrument contain and how many questions were there in each section? Was the questionnaire scored and how was it scored? Without scoring the questionnaire, the authors cannot determine the levels of awareness or knowledge.

Results

Line 167: Change ‘questionnaires’ to ‘questionnaire’ because only one questionnaire was used for the study.

Line 174: Change ‘rare’ to ‘rear’. The authors claimed to have worked with only commercial butchers. How come most of them have become cattle rearers?

Table 1: Change ‘live stock’ to ‘livestock’.

Line 180 – 181. Recast the sentence.

Discussion

The authors seem not to understand that there is a difference between knowledge and awareness, Therefore, it may be worthwhile to define both words in the work. The authors’ claims on knowledge level may not be correct without scoring knowledge to determine the level among the participants. Comparisons made with some author, example Cadmus et al. (2024) do not rhyme.

Limitations

Secondary education is not low level of literacy. Therefore, an area that has more than 70% of the population that attained secondary education cannot be regarded as having low level of literacy.

Reviewer #2: The manuscript “Awareness of Anthrax Disease and the Knowledge of Its Transmission and Symptoms Identification: A Cross-Sectional Study Among Butchers in Ile-Ife” addresses an important public health issue, the awareness and knowledge of anthrax among butchers, that represents a high-risk occupational category in Nigeria. The topic is relevant, timely, also including a One Health perspective. The study provides useful descriptive data that could inform public health education and prevention strategies. The authors present clear findings about the low awareness of anthrax among butchers and the implications for public health. The study is quite good in methodology and analysis, but there are different areas for improvement, before the manuscript could be considered for publication.

As regards the study design, it would be appropriate to clarify the sampling frame, randomization process, and justify the adequacy of the final sample size. How were markets and butchers selected? Was any sampling frame used?

No basic inferential analyses (e.g., chi-square tests) were performed to identify factors associated with awareness and knowledge. Could you please, add this kind of analyses? Despite collecting rich demographic data and having explicit research objectives to assess awareness, no inferential analyses examine whether education, income, or animal sourcing practices associate with knowledge levels. Chi-square tests and logistic regression would identify high-risk subgroups requiring targeted interventions, fulfilling the study's stated aims and enhancing public health utility beyond simple prevalence reporting

The results could be enriched by analysing possible relationships between the respondents' demographics (e.g., education, age) and their awareness or knowledge of anthrax. This would help contextualize the findings.

The discussion could be strengthened by addressing any potential biases in the study (e.g., recall bias due to the use of self-reported data). The paper could explore in greater depth why the awareness was low (e.g., cultural factors, lack of effective public health messaging). While the conclusion addresses the need for improved awareness, it could be more specific about what measures should be taken. For example, the authors could suggest specific communication strategies, community-based interventions, or policy changes.

The figures could benefit from more detailed captions that explain what each figure illustrates beyond just showing the data.

Furthermore, the manuscript contains grammatical errors and spelling mistakes throughout such as “symtoms instead of symptoms in the short title” “Bacillus Anthrax (line 67)” instead of Bacillus anthracis, “unclothed blood (line 83)” instead of “unclotted blood”. Unclotted is better than un-clotted. Line 217 4got? Correct. Write B. anthracis in italics all over the manuscript.

Some sentences are unclear or repetitive, especially in the Background and Discussion sections, so a professional English language editing is recommended.

**Do you want your identity to be public for this peer review?** For information about this choice, including consent withdrawal, please see our Privacy Policy

Reviewer #1: No

Reviewer #2: No

---

## [Decision Letter · Decision Letter 1]

19 Feb 2026

PGPH-D-25-03087R1

AWARENESS OF ANTHRAX DISEASE AND THE KNOWLEDGE OF ITS TRANSMISSION AND SYMPTOMS IDENTIFICATION: A CROSS SECTIONAL STUDY AMONG BUTCHERS IN ILE-IFE

Dear Dr. Adeyemo,

Thank you for your effort in revising this manuscript. After careful consideration, we feel that it has merit but does not fully meet PLOS Global Public Health’s publication criteria as it currently stands. Therefore, we invite you to submit a revised version of the manuscript that addresses the points raised during the review process.

Please find additional suggestions from one reviewer. Kindly revise and address it as soon as possible.

We look forward to receiving your revised manuscript.

Kind regards,

Kanokwan Suwannarong, Ph.D.

Academic Editor

Journal Requirements:

Additional Editor Comments (if provided):

Thank you for your effort in revising this manuscript. Please find additional suggestion from one reviewer. Kindly revise and address it as soon as possible.

Reviewers' comments:

Reviewer's Responses to Questions

**Comments to the Author**

Reviewer #1: (No Response)

Reviewer #2: All comments have been addressed

publication criteria?

Reviewer #1: Yes

Reviewer #2: Yes

3. Has the statistical analysis been performed appropriately and rigorously?

Reviewer #1: No

Reviewer #2: Yes

4. Have the authors made all data underlying the findings in their manuscript fully available (please refer to the Data Availability Statement at the start of the manuscript PDF file)?

Reviewer #1: Yes

Reviewer #2: Yes

5. Is the manuscript presented in an intelligible fashion and written in standard English?

Reviewer #1: Yes

Reviewer #2: Yes

Reviewer #1: Borrowing the authors' words, ' the manuscript has substantially improved, methodologically robust, and

clearly written.' However, there are few issues that need to be addressed as shown in the document attached.

Reviewer #2: I thank the authors for providing the revised version of this manuscript. The authors have made substantial efforts to address the concerns raised in the previous review. Overall, the manuscript has improved in clarity, methodological transparency, and balance of interpretation and I think is now suitable for publication.

**Do you want your identity to be public for this peer review?** For information about this choice, including consent withdrawal, please see our Privacy Policy

Reviewer #1: **Yes:** Akwoba Ogugua

Reviewer #2: No

---

## [Editor Report · Decision Letter 2]

3 Mar 2026

AWARENESS OF ANTHRAX DISEASE AND THE KNOWLEDGE OF ITS TRANSMISSION AND SYMPTOMS IDENTIFICATION: A CROSS SECTIONAL STUDY AMONG BUTCHERS IN ILE-IFE

PGPH-D-25-03087R2

Dear Dr Adeyemo,

We are pleased to inform you that your manuscript 'AWARENESS OF ANTHRAX DISEASE AND THE KNOWLEDGE OF ITS TRANSMISSION AND SYMPTOMS IDENTIFICATION: A CROSS SECTIONAL STUDY AMONG BUTCHERS IN ILE-IFE' has been provisionally accepted for publication in PLOS Global Public Health.

Best regards,

Kanokwan Suwannarong, Ph.D.

Academic Editor